# Stretchable Transparent Light-Emitting Diodes Based on InGaN/GaN Quantum Well Microwires and Carbon Nanotube Films

**DOI:** 10.3390/nano11061503

**Published:** 2021-06-07

**Authors:** Fedor M. Kochetkov, Vladimir Neplokh, Viktoria A. Mastalieva, Sungat Mukhangali, Aleksandr A. Vorob’ev, Aleksandr V. Uvarov, Filipp E. Komissarenko, Dmitry M. Mitin, Akanksha Kapoor, Joel Eymery, Nuño Amador-Mendez, Christophe Durand, Dmitry Krasnikov, Albert G. Nasibulin, Maria Tchernycheva, Ivan S. Mukhin

**Affiliations:** 1Department of Physics, Alferov University, Khlopina 8/3, 194021 St. Petersburg, Russia; vneplox@gmail.com (V.N.); strindberg76@mail.ru (V.A.M.); sungat15004@gmail.com (S.M.); alex.spbau@mail.ru (A.A.V.); lumenlight@mail.ru (A.V.U.); mitindm@mail.ru (D.M.M.); imukhin@spbau.ru (I.S.M.); 2Institute of Machine Engineering, Materials and Transport, Peter the Great St. Petersburg Polytechnic University, Polytechnicheskaya 29, 195251 St. Petersburg, Russia; 3Department of Physics and Engineering, ITMO University, Lomonosova 9, 197101 St. Petersburg, Russia; filipp.komissarenko@metalab.ifmo.ru; 4Nanophysics and Semiconductors Laboratory, University Grenoble Alpes, PHELIQS, IRIG, CEA, 38000 Gre-noble, France; aku.kapoor19@gmail.com (A.K.); christophe.durand@cea.fr (C.D.); 5Nanostructures and Synchrotron Radiation Laboratory University Grenoble Alpes, MEM, IRIG, CEA, 38000 Grenoble, France; aku.kapoor19@gmail.com (A.K.); joel.eymery@cea.fr; 6Centre of Nanosciences and Nanotechnologies, UMR 9001 CNRS, University Paris-Saclay, 10 Boulevard Thomas Gobert, 91120 Palaiseau, France; nuno.amador@c2n.upsaclay.fr (N.A.-M.); maria.tchernycheva@u-psud.fr (M.T.); 7Skolkovo Institute of Science and Technology, Bolshoy Boulevard 30/1, 121205 Moscow, Russia; d.krasnikov@skoltech.ru (D.K.); a.nasibulin@skoltech.ru (A.G.N.); 8School of Chemical Engineering, Aalto University, P.O. Box 16100, FI-00076 Espoo, Finland

**Keywords:** stretchable LED, InGaN, nanowires, single-walled carbon nanotubes, MOVPE

## Abstract

We propose and demonstrate both flexible and stretchable blue light-emitting diodes based on core/shell InGaN/GaN quantum well microwires embedded in polydimethylsiloxane membranes with strain-insensitive transparent electrodes involving single-walled carbon nanotubes. InGaN/GaN core-shell microwires were grown by metal-organic vapor phase epitaxy, encapsulated into a polydimethylsiloxane film, and then released from the growth substrate. The fabricated free-standing membrane of light-emitting diodes with contacts of single-walled carbon nanotube films can stand up to 20% stretching while maintaining efficient operation. Membrane-based LEDs show less than 15% degradation of electroluminescence intensity after 20 cycles of stretching thus opening an avenue for highly deformable inorganic devices.

## 1. Introduction

Stretchable and flexible optoelectronic structures are the envisioned base for the next generation of smart wearable devices with enhanced functionality and usability [1,2,3,4,5,6,7,8,9]. Stretchable optoelectronics is a focus of research for many companies specializing in numerical billboards and displays as a final phase of the evolution of flexible displays. The major advantage is their form-factor adapting for various applications, such as foldable screens [10,11], bio-integrated devices [12,13,14,15,16], and wearable sensors [17,18,19]. Nowadays, the most developed and commercially successful flexible device technology is based on organic materials [20,21]. The main advantage of the organic light-emitting diodes (OLED) is a relatively inexpensive and scalable fabrication coupled with a certain electroluminescence (EL) efficiency. However, organic devices are inferior to inorganic semiconductor emitters in terms of long-term stability, luminance, and external quantum efficiency, especially in the blue and red spectral ranges.

Semiconductor microwires (MWs) have been investigated during the last five years as promising candidates for flexible light emitting diodes [6,22]. Owing to their small footprint, MWs can be bent without structural damage associated with the blue, green, and white emission [23,24]. Thus, integrated polymer/MW membrane withstands stretching and bending with only a negligible deformation of the MWs, thereby preserving the active material integrity. Therefore, a composite polymer/MW membrane can withstand high stretching thanks to the polymer elasticity. Among the earliest demonstrations of polymer/MW (or nanowire) LEDs were ZnO nanowire LEDs [25].

In addition to the stretchable emissive layer, the stretchable LEDs require proper transparent electric contacts. So far, the best stretchable performance in OLED devices was demonstrated with contacts based on four-layer graphene with 80% optical transparency and ~70 Ohm/square sheet resistance [26]. There are also reports describing stretchable OLED devices [27] with Ag (80 nm)/MoO_3_ (3 nm) as anode and Ca (3 nm)/Ag (18 nm) as cathode contacts deposited on photopolymer. The main benefits of this structure are stretchability and flexibility, although the photopolymer surface becomes crumpled after stretching, diminishing device transparency.

Meanwhile, single-walled carbon nanotube (SWCNT) contacts synthesized by chemical vapor deposition have ~20 Ohm per square at 80% transmittance [28] thus providing the best conductivity, transparency, and stretchability compromise. SWCNT contacts on polydimethylsiloxane (PDMS) films preserve the device form due to elastic deformation of the whole structure [29] as the SWCNT films are applied to a pre-stretched PDMS film. A comparison given in [30] of graphene multilayers and CNT stretchable electrodes showed that CNTs are affected less by the deformation than the graphene multilayers (namely, they differ by 1.44 times in terms of conductivity degradation).

In the present study, we combine the two building blocks, both having good flexibility and stretchability, namely the polymer/microwire (PDMS/MW) composite membranes and the transparent SWCNT-based contacts, to demonstrate transparent inorganic stretchable LED. The LEDs are based on InGaN/GaN core–shell microwires producing blue electroluminescence, while the SWCNT films serve as an efficient transparent electrode insensitive to stretching. We show that the electroluminescence spectral shape and intensity are almost insensitive to the stretching conditions for deformations up to 20%. The introduced methods open the route towards stretchable display development based on inorganic light-emitting materials.

## 2. Materials and Methods

### 2.1. Synthesis of InGaN/GaN Microwire Arrays

The GaN MWs with core–shell InGaN/GaN multiple quantum wells (MQWs) were grown by metal-organic vapor phase epitaxy (MOVPE) on sapphire substrates [31,32]. In the first stage, n-doped GaN MWs were grown based on high silane flux that passivates the sidewall surfaces by the spontaneous formation of SiN_x_ ultrathin layer [33]. The GaN MWs are highly n-doped (estimated as ~10^20^ cm^−3^ [34]) with a typical length of 10 μm and a diameter in the range of 0.7–1.5 µm. In the second stage, another non-intentionally doped GaN segment of about 7 μm length was grown without silane flux leading to a reduced doping concentration (about 10^18^ cm^−3^ [35]). The active MQW region was deposited directly on the MW sidewall surface by switching from axial to radial growth thanks to the reduction of the growth temperature from 1040 °C to 750 °C. Due to the presence of the SiN_x_ layer around the wire base, the radial growth is inhibited in the lower wire part and the core/shell heterostructure is only formed around the upper non-intentionally doped wire part, as described in [36,37]. The active region consisted of seven periods of 5-nm InGaN quantum wells delimited by 10-nm-thick GaN barriers and with an indium content of 15%. The growth was pursued with the deposition of the p-GaN 100 nm-thick shell, the hole concentration is estimated to be in the 10^16^–10^17^ cm^−3^ range [38]. The MW density is about 5 × 10^6^ per cm^2^. The MW morphology and the internal structure are illustrated in Figure 1a–c.

The previously published cathodoluminescence (CL) mapping of the InGaN/GaN MWs synthesized under similar growth conditions [39,40] showed that this type of wires is characterized by the main blue emission arising from the radial QWs and an additional long-wavelength emission, which can be associated with a defect band.

### 2.2. PDMS/MW Membrane Fabrication

In this work, the transparent, flexible, and stretchable contacts to MWs were based on SWCNT films. In order to provide an ohmic contact between the SWCNT film and the MW array, 5 nm thick metal layers of Ni/Au were deposited on MW top parts by an e-gun evaporator. Metallization of MW topmost parts was carried out at a 78° deposition angle in relation to the MW axis that allows avoiding undesirable metal deposition on the lower parts of MWs due to the shadowing effect (Figure 1c). After deposition, metalized MWs were annealed in the ambient atmosphere at 500 °C for 10 min to form an ohmic contact to the p-GaN shell [41].

Then, the MW array with pre-deposited thin metal layers was encapsulated into a PDMS layer by G-coating method [42,43]. G-coating is performed with a swinging-bucket centrifuge instead of a standard spinner. To process the samples, we used a high-speed centrifuge Eppendorf 5804.

For the polymer membrane formation, a commercial PDMS (Dow Corning Sylgard 184) was mixed in a base to a curing agent ratio of 10 to 1. The mixture was dropped onto 3 square cm size samples and G-coated at 5000 rpm for 15 min until the sample surface became matt due to the light scattering by revealed MW heads (Figure 1b).

After PDMS deposition, the samples were baked in a muffle oven at 80 °C for 12 h followed by etching in O_2_ plasma to remove the residual PDMS wetting layer from MW top parts for further contact application. The procedure was previously reported in [42] and [43]. Then, the PDMS/MW membranes were released from the growth substrate by a razor blade [42].

### 2.3. Synthesis and Application of SWCNT Contacts

SWCNTs were synthesized by the aerosol chemical vapor deposition (CVD) method described elsewhere [44,45]. SWCNTs were collected directly at the outlet of the reactor by a nitrocellulose filter. SWCNT films used in the present work were of 80% transparency at 550 nm wavelength, while the sheet resistance can reach 20 Ohm/square [28,46].

SWCNT contact pads were applied to free-standing PDMS/MW membranes. In order to fabricate strain-insensitive contacts, SWCNT pads were applied onto pre-stretched PDMS samples: when released at initial non-stretched state nanotube film compresses to form wrinkles and returns to an initial state when stretched [29]. In comparison, SWCNT films applied to non-stretched PDMS demonstrate a significant conductivity degradation after stretching as it was shown in [29] and reproduced in our work further.

It should be noted that the SWCNT films serve as electrical contacts (i.e., transparent electrodes) and are not directly involved in the radiative process occurring in InGaN/GaN MQWs in MWs. Therefore, we consider the SWCNT film wrinkle cavity effect [47] on stretchable LED performance to be negligible. Indeed, in the case of stretchable LED geometry, the light emission is extracted at MW top parts protruding the SWCNT film layer, so the light interaction with SWCNTs is expected to be insignificant due to the film transparency.

As prepared SWCNT films on a nitrocellulose filter were cut to an appropriate size of approximately 0.3 mm^2^, then the contact pads were applied onto both sides of 20% mechanically pre-stretched PDMS/MW membrane and wetted by isopropyl alcohol (IPA) droplets in order to provide an effective adhesion of the SWCNT pads to the PDMS surface. After IPA evaporation, the nitrocellulose filters were easily removed with a soft air flow. In the end, membrane was relaxed and SWCNT contact pads provided a safety margin for subsequent stretches as a stable electrode [29]. The value of 20% mechanical stretch was chosen because of the following reasons: (a) thin PDMS membranes have a high risk of tearing at stretching higher than 30% and (b) 20% is a good compromise between a high transparency and deformability of stretchable SWCNTs (the more a PDMS film is pre-stretched, the less SWCNT film is transparent at normal states due to the wrinkles).

For convenience of electrical measurement, 0.1 mm thin copper wires were applied to SWCNT contact pads with silver lacquer droplets. Then, the whole structure was buried into PDMS except copper wire tails and baked in an oven for 4 h at 80 °C (the workflow is shown in Figure 2).

### 2.4. Optical and Electrical Characterization

Optical transparency of the produced stretchable LED was measured with an integrating sphere. The stretchable LEDs were stuck to glass, which also served as a reference. Argon lamp light was focused on a contacted area of PDMS/MW membrane placed after the integrating sphere slits.

We verified electrical stability of stretchable SWCNT contacts similarly to [29]. The SWCNT stripes were applied to relaxed as well as to pre-stretched by 10% PDMS films (Appendix A). The SWCNT stripes were connected with copper wires and silver lacquer, and then buried into PDMS the same way as the LED device (Figure 2). I–V measurements for reference SWCNT stripes, and for stretchable LED were performed with a Keithley 2400 sourcemeter, and stretching was controlled with a vernier caliper with an accuracy of 0.05 mm. For the stretching test of LED devices, the operating current was fixed, and the device performance was controlled by voltage measurements. EL spectra were measured with an Avantes AvaSpec ULS2048XL spectrometer (Avantes Inc., Louisville, CO, USA).

## 3. Results and Discussion

### 3.1. Device Transparency Measurements

The integral transparency of the stretchable LEDs was estimated as 0.43 compared to the reference Figure 3. This value can be explained by absorption in the two contact layers of stretched by 20% SWCNT ((0.8/1.2) × (0.8/1.2) = 0.44) and in the PDMS/MW membrane (responsible for the remnant 1% decrease). It should be noted that the transmission loss is spectrally flat.

### 3.2. Electrical Characterization

The results of conductivity measurements of the test SWCNT stripes on blank PDMS films demonstrated significantly different rates of degradation depending on the pre-stretching. Specifically, the initially unstretched samples showed 40% of conductivity reduction, while the pre-stretched samples were characterized by only 5% conductivity decrease at 10% stretch (Appendix A). Other pre-stretched rates were thoroughly studied in [29].

In order to prove that the stretchable LEDs keep their electrical properties during stretching, the I–V curves were measured for the as-processed samples, and during stretching cycles and after releasing the strain. The measured I–V curves for a representative device stretched by 20% are presented in Figure 4. A significant current reduction was observed for the first stretching: the current for the relaxed LED is 0.25 mA, while for the stretched LED the value dropped to 0.2 mA at the same voltage of 8 V (a rough estimation of the current density derived from this value gives 10 A/cm^2^, which correlates with the results reported in the literature for thin film InGaN LEDs [48,49]). However, for the subsequent stretching cycles, no further current loss was observed. The knee voltage did not change due to the stretching, indicating a stable SWCNT/MW contact interface. We speculate that the significant decrease of the current at the first stretching can be explained by an impairment of the weakest SWCNT/MW contacts, while the majority of the MWs remained contacted after the initial and consequent stretching. One might suspect that the rigid silver lacquer droplets may introduce local mechanical instability of the contacts at stretching. However, the reference SWCNT contacts did not demonstrate any damages due to the silver lacquer droplets (Appendix A), therefore we conclude that small silver lacquer droplets did not significantly hamper the device stretchability.

Stretching tests confirmed electrical contact stability and the emissive membrane integrity for 20 cycles of stretching (by 10%) and releasing (Figure 4). The voltage was measured for each iteration at a constant LED injection current value of 0.4 mA. The value of stretching was controlled by a vernier caliper with an accuracy of 0.05 mm (i.e., 2.5% of the applied stretching). PDMS/MW membranes showed 15% resistivity increase (deduced from the measured voltage) after 20 cycles under stretched conditions, while in the relaxed state stretchable LED retained the initial value of resistivity. We associate the increase in resistivity with a micro damage of the SWCNT films, which is insignificant at the relaxed state due to wrinkles formation but leads to SWCNT network integrity impairment in the stretched state, when the wrinkles are released, or could be attributed to the baseline stabilization effect observed in a few dozen of the first film stretching cycles (training) [29].

### 3.3. Electroluminescence Measurements

Optoelectronic properties of InGaN/GaN MW arrays, including MW/PDMS membrane configuration, were investigated in detail in our previously reported works including [23,24,31,38,49], etc. Flexibility of InGaN/GaN MW/PDMS membrane LEDs and SWCNT films were also thoroughly studied in [23,24,28,41,42], so it is not addressed in the presented work. Therefore, in this section we focus on stretchability-related properties of the fabricated stretchable LEDs.

EL spectra of a representative stretchable LED were measured at 10 V, 11 V, and 12 V bias applied in the relaxed and 20% stretched states. The relatively high functioning voltage is a typical problem of MW/PDMS LEDs explained by parasitic electric barriers and high series resistance discussed elsewhere [23,24]. As shown in Figure 5, the blue line at 450–460 nm, which originates from MQWs, dominated over the defect yellow band centered at 560 nm [39]. The EL spectra obtained at different biases are vertically shifted for clarity in Figure 5. As the applied voltage increased in the EL experiment, almost no blueshift was observed, which is indeed expected for m-plane QWs [31], which contrary to c-plane QWs do not exhibit polarization internal field screening [50,51]. Under stretching, the EL signal from the contacted area remained homogeneous, indicating stability of MW contacting proven by the preserved blue line intensity.

It should be stressed that stretching did not have any significant effect on the peak positions and spectrum shapes, indicating a high stability of functioning stretchable LEDs. Interestingly, the yellow band became less intense under stretching, especially at lower voltages. This effect will be studied elsewhere.

## 4. Conclusions

In conclusion, we developed and demonstrated stretchable LEDs based on integration of vertical nitride microwire arrays embedded in a PDMS membrane and transparent strain-insensitive SWCNT electrodes. We show repeatable stretching with a minor decrease of the EL signal: less than 15% after 20 cycles. The method for application of SWCNT pads onto a pre-stretch PDMS/MW membrane provides stable stretchable electrical contact. This technology opens new routes for efficient stretchable LED displays and other optoelectronic devices based on inorganic light-emitting materials.

## Figures and Tables

**Figure 1 nanomaterials-11-01503-f001:**
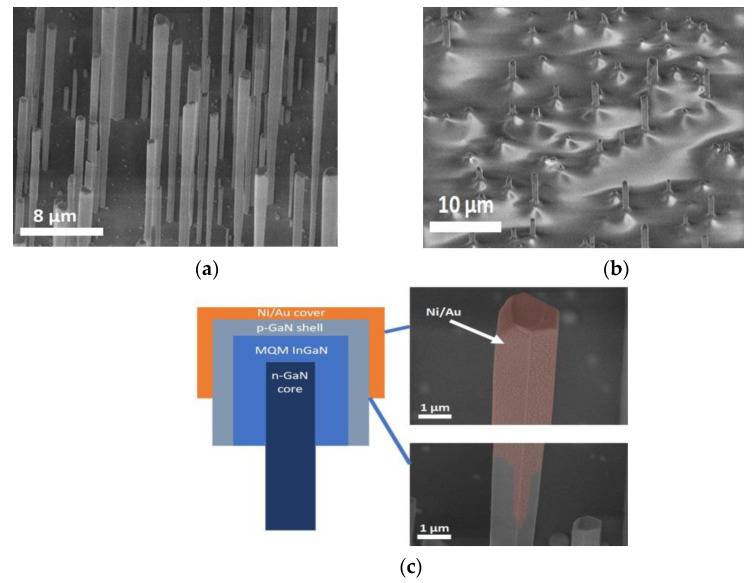
Forty-five-degree tilted SEM images of (**a**) an as-grown MW array, and (**b**) InGaN/GaN microwires encapsulated into PDMS. (**c**) Schematic of InGaN/GaN MQW MW structure with SEM images of a MW tip and the border of an area metalized with Ni/Au (artificial red color denotes metal). The metal layer did not contact MW cores.

**Figure 2 nanomaterials-11-01503-f002:**
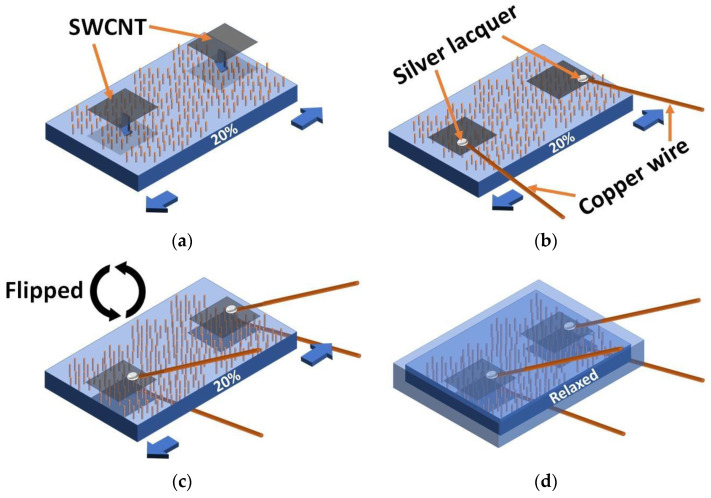
Workflow of electrical contacting to a PDMS/MW membrane: (**a**) application of SWCNT contact pads onto the upper side of 20% pre-stretched PDMS/MW membrane, (**b**) connection to the SWCNT contact pads with copper wires and silver lacquer, (**c**) application of electrical contact to the bottom side of PDMS/MW membrane, (**d**) relaxed LED device buried into PDMS.

**Figure 3 nanomaterials-11-01503-f003:**
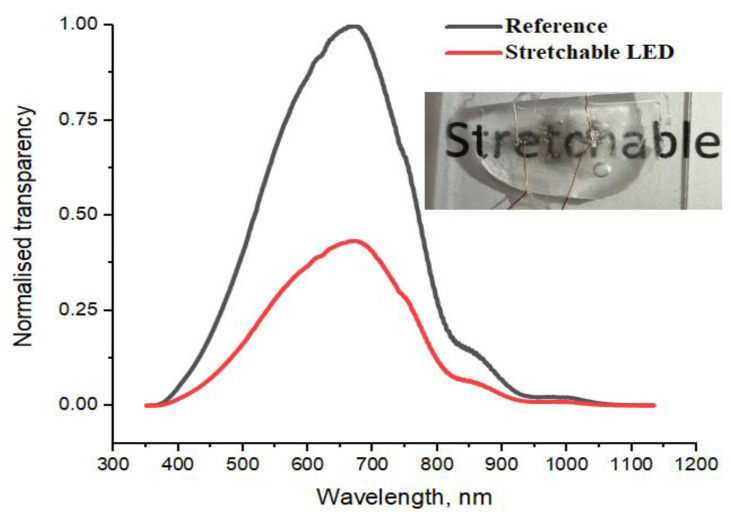
Stretchable LED spectral transparency (**red line**) normalized to the peak value of a reference glass substrate (**black line**). Inset shows a photo of the LED device on reference glass substrate put on a paper with a printed word.

**Figure 4 nanomaterials-11-01503-f004:**
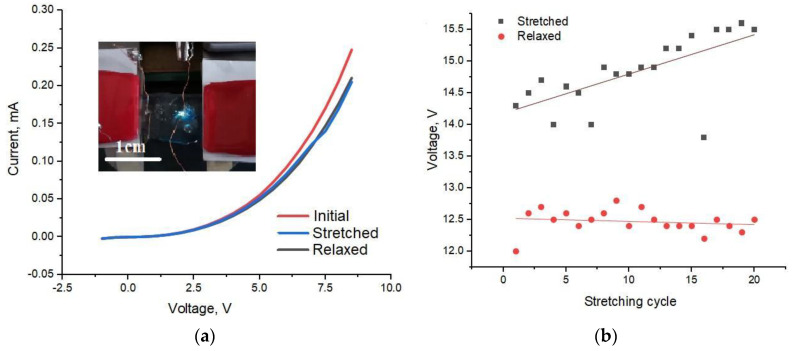
(**a**) I–V curves of a representative stretchable LED in initial (**red line**), stretched by 20% for the first time (blue line), and released (**black line**) states. (**b**) Working voltage during the stretching test of the LED for a constant injection current. Inset in (**a**) demonstrates a photo of the functioning stretchable LED device.

**Figure 5 nanomaterials-11-01503-f005:**
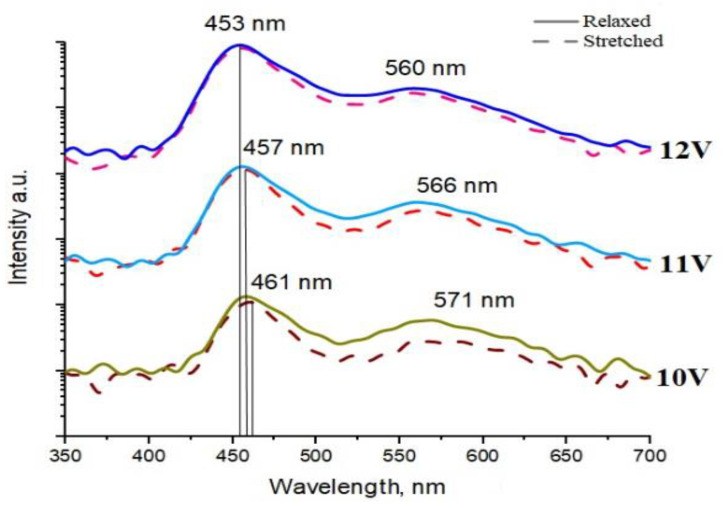
EL spectra of a representative LED in relaxed (**solid line**) and stretched (**dashed line**) states obtained under different applied voltages. The EL spectra are vertically shifted for clarity.

## Data Availability

The data presented in this study are available on request from the corresponding author. The data are not publicly available due to the author’s readiness to provide it on request.

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
