# Peer review of "Stretchable Transparent Light-Emitting Diodes Based on InGaN/GaN Quantum Well Microwires and Carbon Nanotube Films"

_nanomaterials, 2021, doi:10.3390/nano11061503_

Round 1
Reviewer 1 Report
This study combined the InGaN/GaN quantum well microwires (WM) and single-walled carbon nanotube (SWCNT) to obtain the flexible and stretchable inorganic light-emitting diodes. The idea is interesting, this study can be accepted after minor revision.
- Why 20% mechanically stretch is applied on the PDMS/MW membrane before SWCNT films adhesion?
- How to ensure the Ni/Au cover is not covered by the PDMS, providing an ohmic contact between SWCNT film and MW array?
- The EL principle of the SWCNT based MWs should be discussed.
- Why the voltage is increased at the stretched state as the stretching cycle increases?
- The EL performance should be given and discussed, including EQE, current efficiency, and luminance.
Author Response
Dear Reviewer,
Please, find attached the file with answers to all Reviewers.

Reviewer 2 Report
In this work, the flexible and stretchable LED based on core-shell structure InGaN-GaN quantum well microwires/polymer/carbon nanotube assembled device. The quantum-well microwires, polymer hybrid thin films and transparent electrode were well-designed and fabricated. I think this type flexible LED shows promising prospect in flexible photoelectron device applications. The manuscript was well organized, and therefore I recommend it may be accepted for publication only after minor revision.
- The author shows two references about CL research on InGaN/GaN quantum well microwires, but I suggest more detailed research results and discussion about the CL or PL data on microwires/polymer hybrid films, which will be of benefit to understanding the EL spectra in Fig. 5.
- I recommend to evaluating the injection current density of single microwires LED, for that the data of current under different voltage, area of contact electrode, and density of microwires on substrate has been given. So, compare with traditional thin film LED, the injection current density of microwires LED have any different?
Author Response

(The authors gave the same response as above.)

Reviewer 3 Report
The authors presented interesting results of stretchable blue LEDs based on GaN microwires. Especially, Authors claimed that they succeeded in making a relatively stable device by adding a pre-stretch step. However, there were some missing points that the authors should address in their investigation that might be interesting for researchers in this area
1) Experimental results to verify the electrical stability of SWCNT were presented in a supplementary data. However, in this experiment, a different pre-stretched rate (10%) from the main experiment (20%) was chosen. Is there any reason you chose this rate? Also, Have you performed the study of any other pre-stretched rate?
2) In Figure 4, even if the stretching cycle increased, there was no significant change in the relaxed state. On the other hand, the voltage of the stretched state kept increasing. What is the reason for the difference between the two states?
3) The author showed the EL spectra in the relatively high voltage range (10~12V) in Figure 5. Please provide the EL spectra of low voltage region.
4) Authors claimed that they proposed and demonstrated flexible and stretchable devices in the abstract. However, as it is difficult to find the possibility of flexible device in manuscript, it would be necessary to exclude flexible or present new data.
5) You can find flexible nanowire/microwire LEDs since 2011. Update the proper references.
Author Response

(The authors gave the same response as above.)

Reviewer 4 Report
The draft is interesting and deserves to be published after minor but obligatory revision.
What is the architecture and size of the mentioned wrinkle structures that appears in the multilayer films? The mechanism of wrinkle formation and reversibility/irreversibility of the wrinkles as a function of stress should be discussed more. Authors should check the paper (Kolaric & Damman, Applied physics letters, 2010_)
The wrinkle image should be presented in the draft
Also, the authors should discuss s more physical mechanisms that cause different responses as a function of strain especially, the origin and significance of the observed shift in Fig 3 and Figure 5 .
In the end, I am recommending a minor but obligatory revision of this draft.
Author Response

(The authors gave the same response as above.)

Round 2
Reviewer 4 Report
The authors provide correct and detailed answers to all referee questions, and the revised draft reflects that.
Based on that fact, I am recommending the acceptance of the revised draft in unchanged form.